# COVID-19 Maternal Prevention Behavior and Future Intention to Vaccinate for Children

Marjan Zakeri [ID], Ekere J. Essien and Sujit S. Sansgiry *[ID]

Pharmaceutical Health Outcomes and Policy, College of Pharmacy, University of Houston, Houston, TX 77004, USA
* Correspondence: sansgiry@central.uh.edu

**Abstract:** Background: During the COVID-19 pandemic, this study aimed to understand how a mother's current COVID-19 prevention behaviors were associated with the mother's future intention to vaccinate their children against COVID-19. Methods: Using a cross-sectional online survey, mothers who had at least one child between 3 and 15 years old were recruited. COVID-19 prevention behaviors evaluated were wearing a mask, appropriate coughing/sneezing, social distancing, staying home, and handwashing. Participants' age, marital status, race, educational level, incidence of COVID-19 infection in the household, healthcare worker in the household, and future intention to vaccinate children were obtained. Results: Among the 595 participants, 38.3% indicated they did not intend to use the COVID-19 vaccine for their children. Participants with no intention for vaccination had lower mean scores on wearing masks ($p < 0.0001$), social distancing ($p < 0.0001$), staying home ($p < 0.0001$), and handwashing ($p < 0.05$). The incidence of COVID-19 infection in the household was associated with a lower mean score of staying home ($p < 0.01$). Conclusion: Our findings indicate that most mothers were compliant with the CDC recommended guidelines at the time of the survey. Mothers who indicated high adherence to prevention behaviors had a higher likelihood to consider vaccination for their children. Now that the COVID-19 vaccine is available for children as young as six months, healthcare providers need to be aware of the relationship between current prevention behaviors and future intention to vaccinate. They need to counsel parents appropriately with recommendations for children to keep practicing prevention behaviors.

**Keywords:** COVID-19; children; prevention; vaccine; pediatric





## 1. Introduction

Coronavirus disease 2019 (COVID-19) was declared a Public Health Emergency in early 2020, by the World Health Organization [1]. To mitigate the further spread of the disease, the Center for Disease Control and Prevention (CDC) advised preventive measures including staying home, social distancing, and considering good hand hygiene [2]. Many individuals considered the practice of prevention behaviors and reports indicated a high adherence to wearing masks, handwashing, and social distancing early on during the pandemic [3]. A simulated quantitative analysis suggested that prevention behaviors potentially decreased the viral reproduction rate [4]. Another study of 196 countries indicated that the duration of the policy that mandated the practice of wearing masks in public was negatively associated with the mortality rate [5].

The invention of a new vaccine was the primary strategy to combat the COVID-19 pandemic. In December 2020, the Food and Drug Administration (FDA) accepted the emergency use authorization of the first vaccine followed by the national vaccination initiation [6]. While 80–95% of the United States (U.S.) adults had received at least one dose of the COVID-19 vaccine as of October 2022, only 38.6% of 5–11-year-old children had received the first dose of the vaccine by that time. The vaccination rate dropped to 8% for 2–4 years and 5.2% for children younger than 2 years old [7]. The lower rate of vaccination

might be explained by the lower willingness of parents towards the vaccination of their children against COVID-19 [8].

One of the critical steps in proceeding with a healthcare policy to enhance the COVID-19 vaccine uptake is early recognition of individuals' intention to vaccinate. A systematic review study on COVID-19 vaccine hesitancy among United States parents indicated that gender, race, age, level of education, and income are determinants of COVID-19 vaccine uptake [9]. In addition, a recent study indicated that fear of COVID-19 is a predictor of vaccination willingness among adults [10]. Individuals are more willing to vaccinate if more efficacious vaccines with a prolonged protection period are available [11]. However, there is a gap in knowledge about the characteristics of parents, particularly mothers, regarding their willingness to vaccinate their children against COVID-19. In addition, there is a lack of understanding if there is any association between prevention behaviors and future intention to vaccinate the children. Understanding the level of prevention behaviors practiced by those who were infected with COVID-19, and if this experience may affect their decision to vaccinate children, is not clear either. Understanding if practicing prevention behaviors is associated with a higher vaccination rate may help us develop interventions for better vaccine uptake.

The objective of this study was to evaluate an association between mothers' COVID-19 prevention behaviors practiced for their children and the future intention for vaccinating their children. Further, the study evaluated the role COVID-19 prevention behaviors played regarding the COVID-19 infection incidence in the household.

## 2. Materials and Methods

### 2.1. Participants and Study Design

This regional study had a cross-sectional design. We surveyed Texan mothers who were at least 18 years old and had at least one 3–15-year-old child at the time of the study.

### 2.2. Prevention Behaviors

The survey was designed based on a literature review regarding prevention behaviors against COVID-19 [12,13]. Items were modified from their original version to fit the objective of this study. The session on prevention behaviors started with the statement: "In order to prevent Coronavirus infection, I make sure that my child/children". Prevention behaviors were assessed using five items regarding mothers' attention to children's COVID-19 prevention behaviors considered essential by the CDC. The prevention behaviors included wearing a mask (item: "wear a mask whenever they leave home"), appropriate coughing/sneezing (item: "place a tissue paper or bending elbow in front of their mouth and nose when coughing or sneezing"), social distancing (item: "keep a distance of at least 6 feet (2 m) from others"), staying home (item: "do not leave the house unless absolutely necessary"), and handwashing (item: "wash their hands with soap and water before touching anything after entering home"). The frequency of each behavior was measured on a five-point scale where 1 = never, 2 = rarely, 3 = sometimes, 4 = often, and 5 = always.

### 2.3. Intention for COVID-19 Vaccination

The mothers' intentions for considering COVID-19 vaccination for their children were assessed by the mothers' level of agreement to the statement "If the COVID-19 vaccine for children is free and available, I will make sure my child/children get it". The level of agreement was measured on a five-point Likert scale (strongly disagree (1), disagree (2), neutral (3), agree (4), and strongly agree (5)).

### 2.4. Incidence of COVID-19 Infection

To assess the incidence of COVID-19 infection, participants were asked the following question ("did you or anybody you know get infected by the Coronavirus?"). Participants could select several options including "myself", "a member of my household", "A friend or a coworker", and "Not applicable". Participants who selected at least one of the "myself" or

"a member of my household" options were categorized as having a COVID-19 infection in the household. Participants who selected only "A friend or a coworker" were categorized as a COVID-19 infection among friends or coworkers but not the household members. Those indicating "Not applicable" were considered not having a COVID-19 infection in the household or among friends and coworkers.

### 2.5. Healthcare Workers in the Household

Information about the presence of healthcare workers in the household was obtained by one question ("during the COVID-19 pandemic, have you or somebody you know worked as an essential worker (ex., doctor, nurse, paramedic)?"). Participants could select several options including "myself", "a member of my household", "a friend or a coworker", and "not applicable". If participants indicated "myself" or "a member of my household", they were categorized as having a healthcare worker in the household; otherwise, they were categorized as not having a healthcare worker in the household.

### 2.6. Demographic Characteristics

Participant demographic characteristics obtained included age (used as categorical variable), race (White/non-White), marital status (single/separated/divorced/widowed vs. married or in a cohabiting relationship), and the highest level of education (high school diploma/2 years of college/at least 4 years of college).

### 2.7. Procedure

The ease of use, response time, and readability of the survey were evaluated through pilot tests on five mothers who were eligible for the study. Minimal changes were applied to the survey instrument based on the feedback to make it more user friendly. Using the snowball data collection method, the survey was advertised on social media parental groups all around Texas. Social media including Facebook, Instagram, and WhatsApp were searched for parental groups and recruitment was arranged. To collect more data, the Harris County Department of Education distributed the survey link among parents of 3–5-year-old children in the area. Data collection started on 4 March 2021, and ended on 18 March 2021. The survey completion was voluntary and anonymous, with no reward or compensation. Information about the study was provided for individuals and upon their consent, the survey started.

### 2.8. Statistical Analysis

Based on the score for the intention construct, mothers were divided into two groups. Intention scores above 3 were considered as having the intention to vaccinate the child, and a score 3 and below was considered as no intention for vaccination. The decision was based on the fact that mothers who were the focus of this study were those who did not state that they intended to vaccinate their children. We were interested in evaluating all individuals who disagreed with the statement about their intention or those who decided to remain neutral.

The mean score for each preventive behavior was calculated for comparison across the groups. To evaluate the association between the level of prevention behaviors and the chance of COVID-19 infection among the household, participants were recategorized based on having at least one of the household members infected with COVID-19. Participants who indicated themselves or a household member as having been infected with COVID-19 were grouped as COVID-19 infection in the household. While other participants who did not indicate COVID-19 infection among the household were grouped as no COVID-19 infection in the household.

Descriptive statistics were performed for all variables. For the categorical variables, numbers and percentages were reported, while for continuous variables, means and standard deviations were reported. To evaluate the statistical significance of categori-

cal variables between groups, a logistic regression was performed. All statistical tests were considered significant at a *p*-value < 0.05.

*2.9. Ethical Approval*

The study protocol and the survey were evaluated and received approval from the Institutional Review Board of the University. The questionnaire was delivered through the Qualtrics platform. After a brief explanation about the study, consent was obtained from participants.

**3. Results**

*3.1. Participants*

Overall, 1,842 people clicked on the survey link, and based on the inclusion criteria, 595 responses were considered in our study (response rate = 32.3%). Most participants were 36–40 years (35.80%) and White (72.1%). Among the mothers from non-White racial groups, 57 (9.58%) were Asian, 45 (7.56%) were African American, and 64 (10.76%) were from other racial groups. Most respondents were either married or cohabiting (90.25%) and had at least four years of college education (78.15%). In about one-third of households, there was at least one person working in healthcare (36.47%). Most individuals (61.34%) indicated COVID-19 infection only among their friends or coworkers, while 19.16% indicated COVID-19 infection in the household. (Table 1).

**Table 1.** Participants' characteristics in relation with intention to vaccinate their children (n = 595).

| | Total, N (%) | No Intention, N (%) | Intention, N (%) |
|---|---|---|---|
| **Age (years) *** | | | |
| 21–30 | 48 (8.07) | 27 (11.84) | 21 (5.72) |
| 31–35 | 117 (19.66) | 54 (23.68) | 63 (17.17) |
| 36–40 | 213 (35.80) | 74 (32.46) | 139 (37.87) |
| 41–45 | 150 (25.21) | 51 (22.37) | 99 (26.98) |
| >45 | 67 (11.26) | 22 (9.65) | 45 (12.26) |
| **Race** | | | |
| White | 429 (72.1) | 168 (73.68) | 261 (71.12) |
| Non-White | 166 (27.90) | 60 (26.32) | 106 (28.88) |
| **Marital status *** | | | |
| Single/Separated/Divorced/Widowed | 58 (9.75) | 30 (13.16) | 28 (7.63) |
| Married or in a cohabiting relationship | 537 (90.25) | 198 (86.84) | 339 (92.37) |
| **Highest level of education **** | | | |
| High school diploma | 34 (5.71) | 15 (6.58) | 19 (5.18) |
| 2 years of college | 96 (16.13) | 51 (22.37) | 45 (12.26) |
| At least 4 years of college | 465 (78.15) | 162 (71.05) | 303 (82.56) |
| **Healthcare worker among the household** | | | |
| Yes | 217(36.47) | 74 (32.46) | 143 (38.96) |
| No | 378(63.53) | 154 (67.54) | 224 (61.04) |
| **Coronavirus infection among the household members, friends, or coworkers** | | | |
| At least one of the household members | 114 (19.16) | 49 (21.05) | 65 (17.71) |
| Only among friends or coworkers | 365 (61.34) | 131 (57.46) | 234 (63.76) |
| None of the household, friends, or coworkers | 116 (19.50) | 48 (21.49) | 68 (18.53) |

Chi square test results. * *p* < 0.05; ** *p* < 0.01.

*3.2. Intention to Vaccinate*

Overall, 38.32% of mothers indicated a low intention for considering the COVID-19 vaccine for their children. A mother's older age was associated with a higher intention for vaccination ($p < 0.05$). Intention for vaccination was significantly higher among participants who were married or in a cohabiting relationship ($p < 0.05$). Higher education was associated with a higher intention for vaccination ($p < 0.01$). The presence of a healthcare worker in the household had no association with the intention for vaccination ($p = 0.11$). Coronavirus infection among the household members, friends, or coworkers had no association with the intention for vaccination ($p = 0.3$) (Table 1). A further analysis was carried out after regrouping the participants into two groups based on COVID-19 infection in households. We found that participants who experienced COVID-19 infection in the household had similar intentions for vaccination compared to participants who had not experienced COVID-19 infection in their households (57.02% vs. 62.79% $p = 0.25$).

*3.3. Prevention Behaviors and Intention to Vaccinate*

Table 2 provides the mean scores for prevention behaviors in descending order in relation to COVID-19 infection in the household. The most difficult preventive behavior to adhere to was staying home, followed by keeping the 6-feet distance (social distancing). However, the other prevention behaviors were considered relatively easy, including appropriate coughing/sneezing, handwashing, and wearing a mask. Table 3 provides the mean scores for prevention behaviors in descending order in relation to intention to vaccinate. Participants who did not intend to consider the COVID-19 vaccine for their children had a lower mean score on certain prevention behaviors, such as wearing masks ($p < 0.0001$), social distancing ($p < 0.0001$), staying home ($p < 0.0001$), and handwashing ($p < 0.05$). The mean score for appropriate coughing/sneezing was not significantly different across the intention groups ($p = 0.10$) (Table 3).

**Table 2.** Mothers' current behavior in relation to past COVID-19 infection in the household after adjusting for age and education (n = 595).

| COVID-19 Prevention Behaviors | COVID-19 Infection in Household | | |
|---|---|---|---|
| | Total M (SD) | No N = 481(80.84%) M (SD) | Yes N = 114(19.16%) M (SD) |
| Children wearing a mask whenever leaving home | 4.55 (0.92) | 4.57 (0.89) | 4.47 (1.04) |
| Children washing their hands before touching anything after entering home | 4.51 (0.90) | 4.53 (0.89) | 4.43 (0.90) |
| Children placing a tissue paper or bending elbow in front of their mouth and nose when coughing or sneezing | 4.49 (0.89) | 4.47 (0.93) | 4.60 (0.71) |
| Children keeping a distance of at least 6 feet from others | 4.34 (0.86) | 4.36 (0.84) | 4.27 (0.91) |
| Children not leaving the house unless it is absolutely necessary | 3.75 (1.24) | 3.82 (1.21) * | 3.46 (1.33) |

Logistic regression was used for comparison of the mean score of each behavior; the scores on each preventive behavior were from 1 to 5, where the higher number meant a higher level of adherence to the behavior; * $p < 0.05$.

*3.4. COVID-19 Infection in the Household*

Participants who reported COVID-19 infection in their household had a lower mean score of staying home compared to participants who did not experience COVID-19 infection in their household ($p < 0.01$). The mean scores for other prevention behaviors were not significantly different based on the incidence of COVID-19 infection in the household (Table 2).

**Table 3.** Mothers' current behavior in relation to future intention to vaccinate children after adjusting for age and education (n = 595).

| COVID-19 Prevention Behaviors | Total M (SD) | Intention to Vaccinate | |
| --- | --- | --- | --- |
| | | No N = 228(38.32%) M (SD) | Yes N = 367(61.68%) M (SD) |
| Children wearing a mask whenever leaving home | 4.55 (0.92) | 4.24 (1.22) ** | 4.75 (0.59) |
| Children washing their hands before touching anything after entering home | 4.51 (0.90) | 4.38 (1.03) * | 4.59 (0.79) |
| Children placing a tissue paper or bending elbow in front of their mouth and nose when coughing or sneezing | 4.49 (0.89) | 4.45 (0.89) | 4.52 (0.89) |
| Children keeping a distance of at least 6 feet from others | 4.34 (0.86) | 4.07 (1.01) ** | 4.50 (0.70) |
| Children not leaving the house unless it is absolutely necessary | 3.75 (1.24) | 3.42 (1.39) ** | 3.96 (1.10) |

Logistic regression was used for comparison of the mean score of each behavior; the scores on each preventive behavior were from 1 to 5, where the higher number meant a higher level of adherence to the behavior; * $p < 0.05$, ** $p < 0.001$.

## 4. Discussion

Overall, COVID-19 prevention behavior scores were high as indicated by mothers. Our findings indicate most mothers were compliant with the CDC recommended guidelines at the time of the survey. Mothers indicated their children were more compliant with wearing masks, handwashing, and appropriate coughing/sneezing as compared to social distancing and staying home. Further, those who had COVID-19 infection in the household indicated their children did not usually practice the staying home behavior. Not staying home was a significant indicator of COVID-19 infection as compared to the other prevention behaviors. Lastly, those intending to vaccinate their children in the future were practicing better prevention behaviors.

Many mothers indicated that they were intending to consider the COVID-19 vaccine for their children; nonetheless, around 39% had no intention of getting their children vaccinated. Our finding supports the findings of two recent studies on parents' intention for vaccinating their children [8,14]. In the Lazarus et al. study published in the *Journal of Nature Communications*, 42.4% of United States (U.S.) parents were hesitant towards vaccination [14]. The second study on U.S. parents by Hamel et al. found that 25% of parents of children 12 years or beyond definitely would not consider COVID-19 vaccination for their children, 10% of parents would consider COVID-19 vaccination for their children only if it is required, and 18% of parents preferred to wait and see [8]. The variation between our findings and the Hamel study could be explained by the different age of the children in each study and the different design of the questions for evaluating the intention to vaccinate.

On the other hand, parents' lack of intention for vaccination in our study was higher than in another study conducted in Saudi Arabia [15]. Aldakhil et al. found that 21% of mothers were not intending to vaccinate their children, which was dramatically lower than our findings [15]. One of the main reasons explaining the difference in findings is the higher rate of vaccine hesitancy in the U.S. compared to other countries. While about 25% of U.S. adults have vaccine hesitancy, only 10% of adults in Saudi Arabia are hesitant towards vaccination [16–18].

Our study indicated that intention for COVID-19 vaccination was not associated with previous COVID-19 infection in household. However, there was a strong association between the intention for COVID-19 vaccination and higher scores on wearing masks, social distancing, handwashing, and staying home. These findings support our objective that the current prevention behavior practices were associated with the future intention to vaccinate children. While appropriate coughing/sneezing was highly practiced among all children in this study, it was independent from mothers' intention to vaccinate their children.

The high, but similar, scores on appropriate coughing/sneezing among mothers who indicated versus mothers who did not indicate intention regarding COVID-19 vaccination for their children can be explained by the fact that appropriate coughing/sneezing is an old prevention behavior, and this may be highly practiced by people regardless of the COVID-19 pandemic.

Mothers, in general, indicated their children practice a high level of prevention behaviors. The lowest practiced preventive behavior was staying home while the highest was wearing a mask. While mothers paid high attention to all prevention behaviors, staying home and quarantine for children were followed lower than other prevention behaviors. This finding can be explained by the fact that keeping children at home for a long period of time is not an easy task. A recent study during the pandemic showed that only 46% of individuals stayed home when sick [19]. In another study on children and adolescents who were quarantined due to having close contact with an individual who had a positive COVID-19 test, only 16.52% of them had a complete understanding of the rationale for staying at home [20]. In addition, staying home was the only preventive behavior with a significantly lower mean score among the families who experienced COVID-19 infection in their households. Our findings support the findings of a study evaluating the effectiveness of quarantine on the COVID-19 epidemic. They found that a lack of proper quarantine and isolation of infected individuals doubles the infection rate [21].

Aside from other findings, it was positive that most parents ensured their children wore masks and washed their hands. These are probably the prevention behaviors most frequently advertised by the CDC to follow. It should be noted that mothers made sure such behaviors were followed, unlike anecdotal evidence that such behaviors were difficult to follow [20].

Our study also indicates that we may need a consistent policy with respect to prevention behaviors. The changes in the preventive behavior policy by the CDC as well as different policies by each individual state with respect to wearing masks and social distancing may cause confusion and create a perception of deception. These inconsistencies can lead to the complete neglect of prevention behaviors when new variants such as the Delta and Omicron variants turn into the prominent strain. This study provides new insights beneficial for the healthcare providers in the U.S. as well as in other countries to improve the vaccine uptake for children [22].

Now that the COVID-19 vaccine is available for children as young as six months, healthcare providers need to be aware of the relationship between current prevention behaviors and future intention to vaccinate. Healthcare providers should consider that a parent's prevention behavior is a strong predictor of the future intention for vaccination. Therefore, when encountering parents who are not practicing prevention behaviors at an acceptable level, not only can health care provider encourage them to incorporate prevention behaviors, but they can also encourage them to consider COVID-19 vaccinations. They may use different techniques in this regard. A recent study indicated that the most important factor that can be used to increase the intention for vaccination is providing the most accurate and recent scientific information for the parents [23].

Our findings indicated that all mothers, independent from the incidence of COVID-19 infection in the household, were practicing most of the common prevention behaviors for their children. The only prevention behavior that was significantly associated with COVID-19 infection in the household was staying at home. Having this knowledge and other evidence-based information, healthcare providers may counsel parents towards the best decisions for their children.

This study had some limitations. Our sample consisted of mostly White individuals and had a higher average educational level than the general U.S. population, which could potentially affect the generalizability of the study [24]. Based on a recent systematic review, lower levels of education and belonging to the African American racial group are associated with higher vaccine hesitancy [9]. In addition, our sample might not be the best representative sample of all parents, since only mothers were included in this

study. Further, due to the voluntary participation in this study, our results might have been affected by non-respondent bias. The other limitation of our study was the convenient snowball sampling method we used. Using this technique, our sample will likely only partially represent some mothers in society. Furthermore, as with other cross-sectional studies, careful interpretations should be made with the understanding that cross-sectional studies are descriptive, and their goal is to assess the burden of a distinct problem in a target population. It should be noted that this study was conducted before any vaccine for children was available. Understanding vaccine hesitancy when it is available may lead to different results. Another limitation of our study was due to the uncertainty regarding the self-reported information about COVID-19 infection in the household. Some incidences of mild COVID-19 infection might have been not tested and therefore not reported as positive. Lastly, we did not consider the possible effect of immunity via COVID-19 infection on an individual's behavior in the household. Even though we found that not staying at home was associated with a higher chance of COVID-19 infection, there is a chance that mothers who experienced COVID-19 infection in the household changed their behavior regarding staying home because of acquired immunity.

As technology and the use of smart devices increase to obtain rapid information on public behavior, future interventions could be more targeted with appropriate education content specific to groups of individuals who need it the most [25].

## 5. Conclusions

Many mothers practiced and adhered to most COVID-19 prevention behaviors for their children. Current prevention behaviors were highly associated with the future intention to vaccinate. Overall, the association between preventive behaviors and vaccine intentions found in our study can be used to promote and improve COVID-19 vaccination campaigns, targeted messaging, intervention design, and public policy decisions. The findings can be used to educate people about the importance of preventive measures and how vaccination can be seen as a complementary measure to existing preventive behaviors. By crafting targeted messaging for specific groups, such as by identifying groups who are more likely to follow preventive behaviors and have high vaccine intentions, communication can be tailored to reach them more effectively.

**Author Contributions:** M.Z.: Conceptualization, Methodology, Formal Analysis, Data Curation, Data Analysis, Writing—Original Draft. E.J.E.: Conceptualization, Methodology, Writing—Review and Editing. S.S.S.: Supervision, Conceptualization, Writing—Review and Editing. All authors have read and agreed to the published version of the manuscript.

**Funding:** This research received no external funding.

**Institutional Review Board Statement:** The study protocol and the survey were evaluated and received approval from the Institutional Review Board of the University of Houston. (ID: STUDY00002857).

**Informed Consent Statement:** Informed consent was obtained from all subjects involved in the study.

**Data Availability Statement:** The dataset used in this study is available upon request from the corresponding author.

**Conflicts of Interest:** The authors declare no conflict of interest.

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
