# Peer review of "COVID-19 Maternal Prevention Behavior and Future Intention to Vaccinate for Children"

_pediatrrep, doi:10.3390/pediatric15020022_

Round 1

Reviewer 1 Report

Since many social media groups are private, would like an explanation about how the authors gained access to these groups for recruiting purposes. 

The value of 3 on the Likert Scale should be included (Is it neutral?)

Although the sample was predominantly white, would like to see the breakdown of other races.  Also, I don't believe that there were enough non-whites to conclude that there were no differences among racial groups. 

Author Response

Dear Reviewer,

We wanted to thank you for your time and effort in providing constructive feedback. We did our best to respond to all comments.

Please let us know if there is a need for further changes.

Thank you,

Reviewer 2 Report

Thank you for this manuscript. This study that looked at the relationship between COVID-19 prevention behaviours in mothers and vaccination intent. Whereas I think this is an important topic and the work you have done very relevant, I have some concerns about the methodological rigor of this study. 

A couple of concerns arose for me:

1. Introduction: please enhance this by including some of the similar studies done on this same subject, right now the introduction is a bit generic (also need citations for lines 55-61)

2. Materials and methods: Was this survey validated/piloted prior to use? How was the decision reached for the intention scores of 3 and below and over 3? 

3. Results: "Not staying home" was identified as an indicator for COVID-19 infection: however, was the not staying home before the infection or after? Is this not staying home the same as the "children not leaving the house unless absolutely necessary"? Acquisition of covid-19 immunity via infection might have also impacted behaviour.

I would also like to see the data according to child's age (in addition to mom's age). The authors have also not addressed confounders such as: age dependent restrictions based on vaccination status (for those 12 and older). Also, how might have the vaccine availability and Omicron infections and increasing seroprevalence have played a role on the topic? This could be covered in the discussion

A number of weaknesses not addressed:

1. This is mostly White mothers: this is mentioned as a weakness but needs more exploration.  Do White people tend to adhere less/more to recommendations at baseline? need some perspective here

2. Please discuss confounders such as vaccine passports/restrictions based on age of child and how that might have impacted the vaccination intent

3. This is all self report: including whether there was an infection in the household with no corroboration of records (also vaccination intent not corroborated with vaccination uptake)  This weakness needs to be addressed

4. Discussion paragraphs 7 and 8 have are better suited to an opinion piece rather than a scientific paper and need literature corroborations/citations.

If you do revise/resubmit, consider a discussion that is more reflective of your findings (right now it extrapolates quite a bit). I would suggest you try and publish this as a "Brief report" in journal that accepts such publications, rather than a full research article, with a discussion about this paper and some of your results to guide that discussion.

Author Response

(The authors gave the same response as above.)

Reviewer 3 Report

I congratulate with the Authors for the interesting manuscript, in particular I appreciate the intuition to test the relationship between prevention behavior ant intention to vaccinate, it is amazing.

I hope you will appreciate some suggestions to improve the manuscript before publishing. 

TITLE: I know that this may be a strong suggestion, but after reading the manuscript, the perfect title may be 

"Current COVID Maternal Prevention Behavior and Future Intention to Vaccinate for Children2

since the only protagonists are mothers.

You can consider this change, It will confer more appropriateness to the title.

INTRODUCTION

Line 39-41: I am in doubt if this lines are correctly placed. I think that they should be moved after line 60-61, or at least not before have talked about vaccines.

MATERIALS AND METHODS

I strongly suggest to attach in the text an original version of the questionnaire, or all the items with all the multiple choice available answers.

Even if in line 288 Authors declair "this study was conducted before any vaccine for children was available", allover the text no temporary references are present. Please, add the period when the study was conducted.

Another missing theme: when did the Authors stop the answers collecting? 

DISCUSSION

I would like to suggest only an additional take home message thant Authors may insert in discussion section.

From the beginning of COVID Pandemic, the use of online platform for questionnaires and survey administration has strongly implemented the large scale cross-sectional study, since the easiness to provide ther diffusion and the smart fruibility among participants. This is a very actual as well as futuristic concept. It may revolutionize cross-sectional, public-health and tranversal studies/surveys. For this reason, whenever we aim to assess a phenomenon (behaviour, knowledge, etc) we can made up a survey like that and eventually use the results to guide some appropriate interventions (e.g. educational programs, etc). This work assessed behaviors though an online survey, take a look. DOI: 10.19193/0393-6384_2022_3_263 

I appreciate the paragraph on study limitations. I suggest to add further concepts. The snowball data collection method is a peculiar sampling technique, that presents a totally uncontrolled sample selection, since it depends on the first participant, is not random and it depends on the virtual community sharing where you diffused it (in you case). I think that all this aspects may be told.

CONCLUSIONS

This section merits to be improved a bit. 

I hope for your revised version, you're almost there. 

Best Regards

Author Response

(The authors gave the same response as above.)

Round 2

Reviewer 2 Report

Thank you for making the appropriate edits to the manuscript

Please remove lines3320335 in the conclusion (public health vaccine availability decisions need to take into consideration more than vaccination intent)

Author Response

Thank you for the comment. We removed lines 332-335.

Reviewer 3 Report

Dear Authors,

The manuscript is well organized and I would not add anything. 

Good luck for your future research

Best Regards

Author Response

Thank you for all your constructive comments and for helping us improve our work.